


# On the challenges of global entity-aware deep learning models for groundwater level prediction

Benedikt Heudorfer[1], Tanja Liesch[1], and Stefan Broda[2]

[1]Karlsruhe Institute of Technology (KIT), Institute of Applied Geosciences, Kaiserstr. 12, 76131 Karlsruhe, Germany
[2]Federal Institute for Geosciences and Natural Resources (BGR), Wilhelmstr. 25–30, 13593 Berlin, Germany

**Correspondence:** Benedikt Heudorfer (benedikt.heudorfer@kit.edu)

**Abstract.** The application of machine learning (ML) including deep learning models in hydrogeology to model and predict groundwater level in monitoring wells has gained some traction in recent years. By now, the dominant model class is so called single-well models, where one model is trained for each well separately. However, recent developments in neighbouring disciplines including hydrology (rainfall-runoff-modelling) have shown that global models, being able to incorporate data of several wells, may have advantages. These models are often called entity-aware models, as they usually rely on static data to differentiate the entities, i.e. groundwater wells in hydrogeology or catchments in surface hydrology. We test two kinds of static information to characterize the groundwater wells in a global, entity-aware deep learning model setup, first, environmental features that are continuously available and thus theoretically allow spatial generalization (regionalization), and second, time-series features that are derived from the past time series at the respective well. Moreover, we test random integer features as entity information for comparison. We use a published dataset of 108 groundwater wells in Germany, and evaluate the models' performances in terms of Nash-Sutcliffe efficiency (NSE) in an in-sample and an out-of-sample setting, representing temporal and spatial generalization. Our results show, that entity-aware models work well with a mean performance of NSE > 0.8 in an in-sample setting, thus being comparable to, or even outperforming single-well models. However, they do not generalize well spatially in an out-of-sample setting (mean NSE < 0.7, i.e. lower than a global model without entity information). The reason for this potentially lies in the small number of wells in the dataset, which might not be enough to take full advantage of global models. However, also more research is needed to find meaningful static features for ML in hydrogeology.

## 1 Introduction

Groundwater is the primary drinking water resource in Germany (Hoelting and Coldewey, 2013) and a major one worldwide (WWAP, 2015). As such, it is under growing pressure due to e.g. climate change, increased drought frequencies or irrigation (WWAP, 2015). All of these transient drivers of change point to the necessity of functional groundwater management, for which groundwater level prediction models are a key tool. Currently, the dominant hydrogeologic model class – distributed numerical models – are an excellent tool to answer tangible questions in complex hydrogeologic settings. But numerical models do not scale well. It is not trivial to parameterize larger scale numerical models because it is expensive. The process needs some degree of subjective sophistication and depends on amounts of unstructured geological or spatial data. The ability of machine learning



algorithms, especially neural networks, to incorporate large amounts of data and letting them sort out the complexity themselves makes them a useful tool to close this gap. With large-scale applicability also comes the potential to realize Prediction in Ungauged Aquifers (PUA), as put forward by Heudorfer et al. (2019) and Haaf et al. (2020, 2023), in a global, generalizable way. The idea of PUA is inspired by the widely known concept of Prediction in Ungauged Basins (PUB) in rainfall-runoff hydrology (Sivapalan et al., 2003; Hrachowitz et al., 2013). As the name tells, it refers to the aim of setting up models that

are capable to predict groundwater levels in areas where no groundwater data is sampled, i.e. ungauged areas. This can be large areas with no or sparse monitoring, but also areas in between groundwater monitoring wells in well-monitored regions. As groundwater dynamics can change within rather short distances due to heterogenous aquifer or infiltration conditions, such models could be used to create spatially continuous data out of point measurements at monitoring sites. Continuous groundwater level data are of great importance, as they serve as a basis for important decision-making tasks, such as deriving

protection zones.

Even though the application of machine learning approaches in hydrogeology gained some traction in recent years, the field is still emerging. In a recent review, Tao et al. (2022) recapitulated that neural network architectures often prove to be superior to other machine learning model classes. Thereby, state of the art groundwater level prediction with neural networks is mostly based on single-well models, where a single neural network model is trained for each groundwater well (Tao et al., 2022;

Wunsch et al., 2021b, 2022a; Rajaee et al., 2019). While excellent fits can be achieved like this on the level of individual wells, the big drawback is that it is not possible to generalize or even regionalize with these models. A model that is capable to overcome this drawback is called a "global model", where "global" means encompassing the whole dataset available. This model class made a first appearance in hydrogeology's sister field, hydrology, by the works of Kratzert et al. (2018, 2019a, b) on the task of rainfall-runoff modelling. They showed (Kratzert et al., 2019b) that a neural network can use static features (like

time series features or environmental features, see below) to distinguish individual dynamic input features (meteorological) and target features (there: runoff) relations in a meaningful way, thereby allowing the model to generalize to locations with similar combinations of static features. They called this setup an "entity-aware model" (Kratzert et al., 2019b), a term which is also common among various disciplines that use entity characteristics to model personalized prediction of responses for individual entities caused by external drivers (Ghosh et al., 2023).

While there have been approaches of global models for groundwater level modelling (e.g. Clark et al., 2022), to the best of the authors' knowledge, the concept of entity-aware global modelling has not been transferred to hydrogeology up to date. The reason might be that providing the model with a set of static features that is able to capture the hydrogeologic dynamics of the system in a conceptual way is much more difficult than in other disciplines, due to the complex nature of underground flow. Even though hydrology and hydrogeology are strongly related disciplines (Barthel, 2014), the question is whether the global

entity-aware approach will work similarly well in hydrogeological conditions, which exhibit substantially more heterogeneous and "local" groundwater dynamics (Heudorfer et al., 2019), which is why nearby wells do not necessarily have high time series similarity (Wunsch et al., 2022b). It is not trivial to expressively link environmental static features taken from geospatial or geologic data products to groundwater dynamics (Haaf et al., 2020), also because of uncertainties in these products, which are often also only regionalized data from point measurements, such as for hydraulic conductivity or depth to groundwater.





For this reason, we test two different versions of a hydrogeologic global entity-aware model: the time series feature driven model (TSfeat model) and the environmental feature driven model (ENVfeat model). Both models have different potential applications. The ENVfeat model uses static environmental features that are available spatially and continuously (see section 2.2 for more information). It represents the gold standard of a fully generalizable and regionalizable model that we seek to achieve, in order to reach the overarching goal of PUA. However, geospatial data availability of sufficient quality to be used

in machine learning is not always available. Also, above-mentioned lack of representativity of geospatial proxy data with regard to groundwater dynamics (Haaf et al., 2020) hypothetically hampers predictability in a hydrogeologic ENVfeat model. Thus, the TSfeat model is the viable alternative that retains the property of generalizability. The TSfeat model differentiates groundwater time series in multi-well prediction based on static time series features that are derived from the past groundwater time series itself (Heudorfer et al., 2019; Wunsch et al., 2022b, , see section 2.2). A TSfeat model can therefore hypothetically

work on any existing monitoring network, being able to incorporate additional unseen groundwater data and thus generalize on groundwater data alone, despite lacking the ability to regionalize in the sense that it cannot predict groundwater head based on secondary data sources (i.e. geospatial data) like the ENVfeat model. We argue that time series features are best suited to describe the dynamics of an individual time series, and to best distinguish different groundwater dynamics from each other, thus hypothesizing that the TSfeat model outperforms the ENVfeat model.

In general, both "global" model types have the theoretical advantage of more data compared to single-well models, i.e. data of similar wells regarding dynamics can contribute to the training of e.g. wells with few training data and thus enhance their performance. To further test our hypotheses, we compare additional model setups, one where the set of environmental or time serie static features are replaced by a set of completely random static features (RNDfeat models), and one with no static features at all, only relying on the dynamic inputs (DYNonly model).

Regarding architecture, we use Long Short-Term Memory neural networks (LSTM, Hochreiter and Schmidhuber, 1997) in the global model setup to learn the input-output relationship of meteorological forcing data and groundwater level, and combine it with a Multi-Layer Perceptron (MLP) for processing the static features. This is despite Wunsch et al. (2021b, 2022a) resorting to the Convolutional Neural Network (CNN) architecture for the single well setup because of better performance and stability. The reason is that LSTM showed overall better performance in preliminary experiments to this study (see figure A1). LSTMs

are an improved version of simple Recurrent Neural Networks (RNN) and overcome RNNs drawback of limited memory of only few time steps (Bengio et al., 1994) which makes LSTMs suitable for time series modelling. It does so by introducing a cell memory state as well as an input gate, forget gate and output gate to control information storage and dissipation when flowing through the LSTM layer during training (Hochreiter and Schmidhuber, 1997). LSTMs are a common architecture for sequence modelling, e.g. in hydrological settings (Kratzert et al., 2018, 2019a, b), and increasingly in groundwater (Tao et al.,

2022; Rajaee et al., 2019).

The aim of this study is to test whether the concept of entity-aware global deep learning modeling is transferable to groundwater level prediction, and, if so, which set of static features are best suited to do so. First, we introduce the dynamic (section 2.1) and static (section 2.2) feature data used in the study. We then give an overview of the model architecture and optimization strategy (section 3.1) and an outline of the experimental design (section 3.3), followed by a brief introduce of the state-of-



the-art learning rate scheduling method to help learning (section 3.2). We then compare model performance of the ENVfeat, TSfeat, RNDfeat and DYNonly model variants (section 4.1) and discuss possible reasons. We further analyse feature importance (section 4.2) to get to the bottom of performance differences. The paper ends with concluding remarks and an outlook in section 5.

## 2 Data

### 2.1 Dynamic feature data

Regarding groundwater level data, the dataset of Wunsch et al. (2022a) is used in this study. This dataset consists of 118 weekly groundwater level time series from the uppermost, unconfined aquifer layer, which groundwater dynamics are mainly dominated by climate forcing. The wells are distributed across Germany and were picked to make the spatial coverage as representative as possible within the national borders. Also, only wells that proved high forecasting accuracy in past experiments were used in this dataset. In addition, a low percentage of gaps was filled with appropriate interpolation tools. For additional
information on the dataset and details on data preprocessing routines please refer to Wunsch et al. (2022a). Based on this readily available dataset, the only additional preprocessing step was to exclude time series with start dates after 01.01.2000 and end dates before 31.12.2015. This was done to make sure that enough data is present for good model results. Also, four individual time series were excluded manually because of missing environmental static feature data (see section 2.2). This resulted in a total number of 108 groundwater time series used in this study. Dynamic input features used in this study are precipitation
(P), temperature (T) and relative humidity (rH) from HYRAS 3.0 (Rauthe et al., 2013; Frick et al., 2014), as well as an annual sinusoidal curve fitted to temperature (Tsin), which proved to be a valuable driving variable (Wunsch et al., 2021b). HYRAS 3.0 is a 1x1 km gridded meteorological dataset covering German national territory, with data ranging from 1951 to 2015. The dataset is essentially the same as in Wunsch et al. (2022a), however HYRAS' relative humidity and the fitted sinusoidal curve was used here additionally.

### 2.2 Static feature data

It is well known that no single static feature is able to describe the totality of control on groundwater dynamics, but a combination of features can provide a good approximation (Heudorfer et al., 2019). Consequently, exhaustive yet compact sets of static input features were compiled. Thereby, two separate sets of static features were fed into two separate model setups:
time series features and environmental features. The first type (time series features) are quantitative metrics calculated from the groundwater time series themselves and express certain aspects of dynamics in these time series. There is a long history of studies in hydrology and to a somewhat lesser extent in hydrogeology (see introduction of (Heudorfer et al., 2019) for a brief review) dedicated to finding, improving and analyzing which time series features best depict certain aspects of time series dynamics, or the totality of time series dynamics in a reduced set of time series features. Oftentimes redundancy analyses,
correlation analyses, dimensionality reduction or similar methods are conducted to determine a suitable set of features. As a





**Table 1.** Static time series features used in the TSfeat model variant. Features were derived from the groundwater level (see section 2.1) from the beginning of the time series until 2011 (to exclude the test period from 2012 onwards). Table taken from Wunsch et al. (2022b).

| short name | feature name | description | citation |
|---|---|---|---|
| RR | Range Ratio | Detection of superimposed long-periodic signals, also sensitive to outliers, calculated as the ratio of the mean annual range to the overall range | Wunsch et al. (2022b) |
| Skew | Skewness | Boundedness, inhomogeneities, outliers, asymmetry of the probability distribution | Wunsch et al. (2022b) |
| P52 | Annual Periodicity | Strength of the annual cycle, calculated by correlating (Pearson) the mean annual (52 weeks) periodicity with the complete time series | Wunsch et al. (2022b) |
| SDdiff | SDdiff | Flashiness, frequency, and rapidity of short-term changes, calculated as the standard deviation of all first derivatives | Wunsch et al. (2022b) |
| LRec | Longest Recession | (unnaturally) long descending heads, longest sequence without rising head values | Wunsch et al. (2022b) |
| jumps | Jumps | Inhomogeneities/breaks, partly also variability, calculated as the absolute and standardized maximum change of the mean of two successive years | Wunsch et al. (2022b) |
| SB | Seasonal Behaviour | Position of the maximum in the annual cycle, agreement with the expected average seasonality (Min in September, Max in March) | Wunsch et al. (2022b) |
| med01 | Median[0,1] | Boundedness, median after scaling to [0,1] | Heudorfer et al. (2019) |
| HPD | High Pulse Duration | Average duration of heads exceeding the 80th percentile of non-exceedance | Richter et al. (1996) |

conceptual decision, we use the set of time series features devised by Wunsch et al. (2022b). This set constitutes a small and manageable, redundancy-reduced set of time series features that was furthermore successfully applied in past studies to cluster time series in the larger dataset of wells (Wunsch and Liesch, 2020; Wunsch et al., 2022b) from which the sample of wells used in this study are taken (see section 2.1). The full list of time series features used in this study, along with descriptions, can be found in table 1.

The second type of features, environmental features, are descriptors of the hydrogeological, physiographic and climatic functioning of the underground and landscape. They are proxies for environmental factors controlling groundwater recharge and flow and thus the dynamics in groundwater time series (Haaf et al., 2020). To be able to reach the stated goal of PUA (see section 1), it is important that environmental features used in the model are spatially continuously available across the study domain (Germany). Thus, only nationwide geospatial datasets are considered in the selection which are, for the sake of





**Table 2.** Environmental static features used in the ENVfeat model variant. Hydrogeologic, soil, topographic and land cover features as well as ETPpot were derived from map products. Climatic static features with exception of ETPpot were derived from the meteorologic dynamic input features (see section 2.1) in the period from 1991 (to match ETPpot data availability) until 2011 (to exclude the test period from 2012 onwards).

| Type | Short name | Description | Unit | Citation |
|---|---|---|---|---|
| Hydrogeologic | Recharge | Mean annual groundwater recharge rates 1961-1990 | mm | BGR (2019) |
| | Percolation | Mean annual groundwater percolation rates 1961-1990 | mm | BGR (2003) |
| | Hygeo_division | Divisions/areas of similar hydrogeologic properties, groundwater conditions and geologic genesis | categ. | BGR & SGD (2015) |
| | Conductivity | Hydraulic conductivity of the aquifer | m/s | BGR & SGD (2019) |
| | Aquifer type | Classes of aquifer types (porous, karstic, fractured, mixed) | categ. | BGR & SGD (2019) |
| Soil | Soil type | Soil type | categ. | BGR (2018) |
| | Clay | Clay content of the soil in weight fraction | % | Hengl et al. (2017) |
| | Sand | Sand content of the soil in weight fraction | % | Hengl et al. (2017) |
| Topographic/ drainage | TWI | Topographic Wetness Index | — | calculated after Beven and Kirkby (1979) |
| | Divide to stream | Distance from hypothetical groundwater catchment divide to nearest stream (hydrologic order 3) at the groundwater well location | m | Noelscher et al. (2022) |
| | Lateral position | Relative position of the groundwater well lateral along the divide-to-stream stretch (hydrologic order 3) | — | Noelscher et al. (2022) |
| | Stream distance | Distance from the groundwater well to nearest stream (hydrologic order 3) | m | Noelscher et al. (2022) |
| Land cover | CLC land cover | CORINE land cover classes (5 ha scale) | categ. | EEA (2018) |
| Climatic | Tmean | Mean annual average temperature | °C | self-derived |
| | Psum | Mean annual sum of precipitation | mm | self-derived |
| | ETPpot | Mean annual sum of potential evapotranspiration | mm | DWD Climate Data Center (CDC) (2023) |
| | rHmean | Mean annual average air humidity | % | self-derived |
| | Frostdays | Mean annual percentage of days frost days (< 0°C) | % | self-derived |





reproducibility, freely available. Moreover, we use datasets that are not too fine-grained (in the case of categorical data), in order to ensure that each category is represented by one or more monitoring wells.

In surface hydrology, the science of identifying major controlling factors for river flow systems is mature enough to yield large-scale or even global selection datasets of environmental features with which river flow can be predicted and explained in
an exhaustive way (e.g. Addor et al., 2017; Linke et al., 2019; Kratzert et al., 2019b, 2023). In hydrogeology, this is not the case yet. No comparable selection of tried and tested sets of environmental features controlling groundwater dynamics is available that could be used in machine learning based prediction of groundwater head. In light of that, we compiled a first selection that is summarized in table 2. It was assembled primarily by conceptual decision from available geospatial datasets to cover the five major domains of control, namely hydrogeology, soil, topographic drainage, land cover and climate. Some factors such as e.g.
depth to groundwater, which are important factors based on conceptional understanding, are omitted, due to above-mentioned data availability reasons. Thus, this selection should be seen as a starting point to serve the proof of concept in this study.

For testing our hypothesis, a third set of static features was used, namely features with completely random integers. This was done in two variants, first, with a number of nine random features to equal the number of time series features (RNDfeat9), and second with 18 random features to equal the number of environmental series features (RNDfeat18). These numbers were
chosen to make sure that all aspects but the values of the static features themselves match the model setup of the TSfeat and the ENVfeat models to exclude other influences.

## 3 Methods

### 3.1 Model architecture and optimization strategy

We use LSTMs in a global entity-aware model setup to learn the input-output relationship between dynamic (meteorological)
input features and groundwater level as the target feature. LSTM was chosen above CNN because it proved to be superior to CNN in the global entity-aware model setup in preliminary studies (see figure A1 for a comparison). To allow the global model to distinguish between different groundwater dynamics of individual wells, static input features that differentiate the wells must be fused to the dynamic (meteorological) input features. Different approaches exist to accomplish this data fusion. Most notably, Kratzert et al. (2019a, b) provide two separate variants to accomplish data fusion. The first is the basic variant where
static features are simply concatenated to the meteorological inputs at each time step, together entering the model through the same input layer. The second is more sophisticated with a modified LSTM layer where static features control the input gate, and the dynamic features control the forget gate, output gate and memory. Even though the modified LSTM layer variant provides desirable levels of interpretability, the basic model variant notably outperformed it (Kratzert et al., 2019b). Thus, we disregard the modified LSTM layer variant in this study. However, instead of the oversimplistic design of Kratzert's basic variant, we
present a model architecture where data fusion of dynamic and static input features is achieved by providing separate model threads that process the dynamic and static inputs individually and are later concatenated (fig. 1). For every time step being processed by the model, a sequence of the previous 52 time steps (making up one full year) of the four dynamic input features P, T, Tsin and rH is given to a LSTM layer of size 128 in the dynamic model thread. In the same time step, one set of static feature





values, associated with the well whose groundwater head is currently being processed, is fed into a Multi-Layer Perceptron
(MLP), with one fully connected (Dense) layer of size 128 in the static model thread. Subsequently, outputs of both threads are
concatenated and fed into another MLP with a Dense layer of size 256, which again feeds into an output Dense layer of size
one. In the whole architecture, all neural layers (despite the output) are followed by a dropout layer with a dropout rate of 0.1
for regularization.

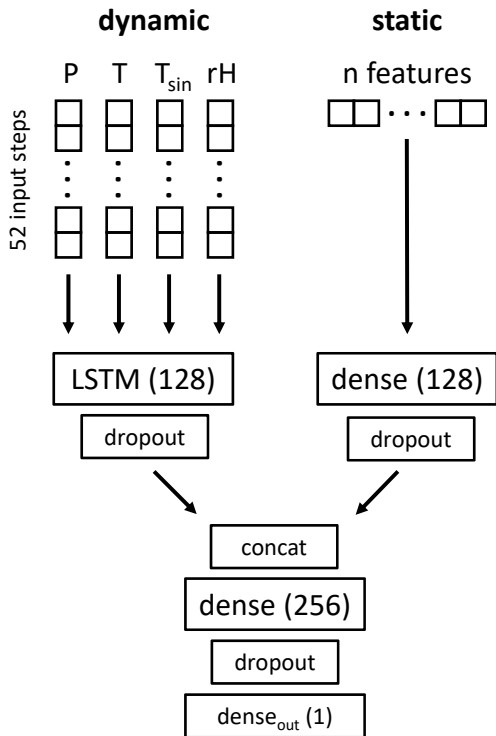

**Figure 1.** The double threaded global entity-aware model architecture introduced in this paper. Hyperparameters not found in the figure are
reported in the text or can be found in the associated code.

For every well separately, both groundwater and meteorological time series were split into three parts: Training set, validation
set and test set. To ensure comparability of performance between wells in light of interannual (or even interdecadal) fluctuations
in groundwater, we chose to set a fixed time period for the validation and test sets. The validation period was scheduled
01.01.2008 - 31.12.2011, and the test period was scheduled 01.01.2012 - 31.12.2015, which is four years each. The training
period was left open towards the past, meaning the model takes various time series lengths during training, under the sole
condition that training data is available at least from 2000 onwards, i.e. at least eight years of training data (see section 2.1).
The model was optimized with the Adam optimizer (Kingma and Ba, 2014) on the Mean Squared Error (MSE) loss function
during training, and later evaluated based on Nash-Sutcliffe Efficiency (NSE) calculated from model errors in the test set. As
Kratzert et al. (2019b) rightfully noted, MSE and NSE are both squared error loss functions, with the difference that the NSE





is normalized by variance. This implies that MSE and NSE are linearly correlated and will yield the same model results in a single well model setup. However, in multi-well model setups this linear relationship is lost because of differing mean and

variance of observed groundwater levels in different wells, heavily altering un-normalized MSE scores. As a remedy, Kratzert et al. (2019b) introduces a custom, basin-averaged NSE, where during training NSE loss is calculated per basis and averaged afterwards. We applied a simpler solution with the same effect, by standardizing each groundwater time series separately during preprocessing. This way, the linear relationship between MSE and NSE is restored in the multi-well model setup when using MSE as a loss function during training and NSE as an evaluation metric, while at the same time avoiding the use of

computationally expensive custom loss functions.

### 3.2 Learning rate scheduling

To avoid rapid overfitting and exploding gradients, a behavior not uncommon in LSTM models (e.g. Goodfellow et al. (2016)), we used a relatively large batch size of 512 (3 permille of the 160,415 samples in the training set) to make the learning process less stochastic and thus more stable. Moreover, we decreased the overall learning rate to 0.0003 (from the Keras default of

0.001). To further improve learning efficiency, we applied a learning rate schedule (LRS) combining a learning rate warmup with subsequent learning rate decay. Warmup is a limited phase in the beginning of training where the learning rate is gradually increased until it reaches a target learning rate ($lr_{target}$). This fights early overfitting by reducing the primacy effect of the first training examples learned by the model, since in unbiased datasets, the model can learn "superstitions" from the first learning examples otherwise uncommon in the dataset. We use a warmup period of 1 epoch, starting from $lr_0 = 0$ to the above-mentioned

$lr_{target}$=0.0003. After the initially high $lr_{target}$ is reached, it is slowly reduced again. This strategy - warmup periods follod by initial high learning rates - was shown to improve the performance of neural networks (Smith and Topin, 2018; Li et al., 2019). In our case, it led to a strong stabilization of the loss curve (see fig. 2) at unchanged performance. We used a cosine shaped learning rate decay after warmup using the formula below, slightly differing from ready-to-use implementations e.g. in tensorflow.

$$lr_i = 0.5 * lr_{target}(1 + cos(\frac{\pi * (i_i + i_{warmup})}{i_{total} - warmup})) \tag{1}$$

Where $lr_i$ is the learning rate at the current batch step, $lr_{target}$ is 0.0003, $i_i$ is the integer of the current batch step, $i_{warmup}$ is the number of batch steps during warmup (number of total training samples divided by batch size) and $i_{total}$ is the total number of batch steps during all training epochs. In our case, training spanned 30 epochs. The shape of the LRS is illustrated in figure A2.

Figure 2 shows the loss curves for 10 seeds with and without the LRS for the in-sample ENVfeat and TSfeat models (see figure A3 for loss curves of the other model variants). The TSfeat model does have a slight MSE improvement with a mean of 0.002 when using LRS, but the ENVfeat model actually performs slightly worse with a mean MSE decrease of 0.008 when using LRS. The same can be said when considering NSE in the test period, where mean performance decreases by 0.0005 with LRS. These changes in performance are negligibly small fluctuations around zero, and in essence, it can be said that



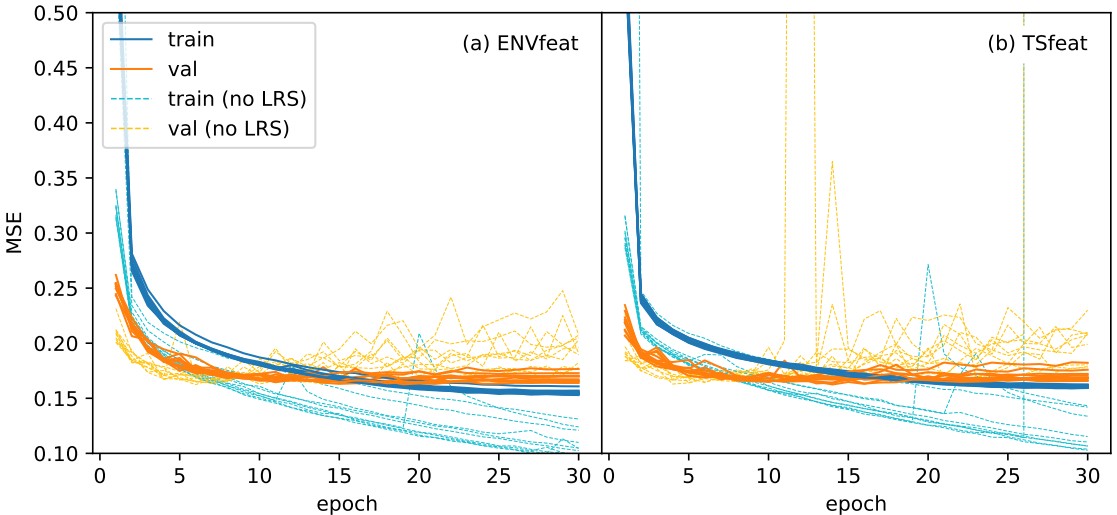

**Figure 2.** Losscurves of the training and validation period for the 10 different seed initialization runs for the (a) ENVfeat model and (b) TSfeat model, each with and without the learning rate scheduling described in section 3.1. The figure shows a strong stabilization effect on losscurves.

model performance is basically the same with or without LRS. Thus, no greatly improved performance can be observed, as was achieved by e.g. Smith and Topin (2018) and Li et al. (2019). However, introducing LRS comes with a strong stabilization effect on the loss curve (figure 2). While learning without LRS shows tendencies of rapid overfitting as well as to heavily exploding gradients, none of that is visible when using LRS. Loss curves with LRS are near-ideal, with train and validation loss curves approaching each other gently, and validation loss curve never significantly increasing after reaching its minimum.

The greatly increased stability of the loss curve implies much better generalization abilities of the model when applying LRS and shows the big advantage of this technique.

### 3.3   Experimental design

After evaluating the effect of the LRS approach on training, we set up three different global model variants with static features. Using the model architecture described in section 3.1, we either used static time series features, static environmental features

or random static features in the static model thread to build the time series feature model (TSfeat model), the environmental feature model (ENVfeat model), and the random feature model (RNDfeat model) respectively. The RNDmodel was run with two different numbers of features, i.e. 9 to be consistent with the number of features in the TSfeat model, and 18 as used in the ENVmodel. Additionally, we run a ground truthed model variant without any static features, i.e. only the dynamic strand (DYNonly model) as described in section 3.1.

To test the models' performance, we first run all models in an in-sample (IS) setting where all wells are used for training and performance is evaluated for each well separately based on the NSE score in the test period. We then compare their test score





(NSE) to the results of the single station models of Wunsch et al. (2022a), i.e. models trained and hyperparameter-optimized on every well separately. Importantly, we took their published results for this and did not rerun any of the single station models. Also, it has to be noted that comparing with the single station scores of Wunsch et al. (2022a) is not benchmarking in a narrow
sense, since some differences beyond the model architecture exist, e.g. Wunsch et al. (2022a) did not use rH as an input (which is used here) and optimized the input sequence length (which is fixed to 52 here).

To further test desired capabilities of spatial generalization, all global models were additionally run in an out-of-sample (OOS) setting, where the models are tested on unseen data, i.e. wells not used for training. To be precise, for every well the models are trained leaving the wells' entire data out of training, and then predicting the wells' score using only the wells' test
data. Practically, this would equate to applying a Leave-One-Out (LOO) cross validation, but due to computational constraints, we used 10-fold cross validation instead to test the described out-of-sample performance. To ensure robustness of the results, all models were run with 10 different seeds for random weight initialization on both settings (in-sample and out-of-sample).

Finally, to understand the inner workings of the model, and how its performance relates to its input data, Permutation Feature Importance (Breiman, 2001; Fisher et al., 2019) was applied. Thereby, we measure the importance of features by successively
taking every individual input feature (dynamic or static) of the trained model, permuting it by shuffling it randomly, and then calculating the models' error (here: MSE) in the test data. A strong increase in the models' error equates to a high importance of the feature being shuffled. This was repeated for each of the 10 randomized initialization seeds.

An additional side experiment was to compare the LSTM-based model to a modified version where the LSTM layer is replaced by a CNN layer, as suggested by the results of (Wunsch et al., 2021b). However, the results of the CNN variants
showed consistently lower performance in the used model setup (see fig. A1), so the results shown in the following focus on the ones achieved with the LSTM models.

## 4 Results and Discussion

### 4.1 Performance comparison of model variants

A side-by-side comparison of all global model scores and the single-well model scores of Wunsch et al. (2022a) can be seen
in figure 3. The mean, lower (10%) and upper (90%)-percentile NSE of the 10 ensemble members for all model variants are shown in table 3.

In the in-sample validation (IS), all global models with the exception of the DYNonly model perform almost identically with only minor differences, at the level of statistical noise. Only the RNDfeat9 model seems to show a slightly lower performance. Two things are somewhat unexpected in this result, first, that the TSmodel does not outperform the ENVmodel, and second
and even more striking, that the RNDfeat models can keep up with the performance of the models with "meaningful" static features. This result corresponds to the findings of Li et al. (2022a), who replaced their static environmental features with random counterparts and found similar or even improved performance for rainfall-runoff modelling in CAMELS-basins. We speculate that the models use the static features solely as a kind of "unique identifier" for the wells, thus, it doesn't seem to matter if the static values represent some meaningful information (in terms of generalization) or not. This shows that our model





is not able to learn from wells with similar static features, probably due to the number and choice of wells in the considered dataset. The reason for the inferior IS performance of the DYNonly model however seems obvious: since no static features are provided, the model is not able to distinguish between the different wells and rather fits to some average behaviour of all wells.

All global models with static features also slightly beat the scores of the single-well models. This result confirms observations by Kratzert et al. (2019a, b), who also observe better scores of global models over single-station models in rainfall-runoff
prediction. This seemingly contradictory results – after all, single-well models are specifically optimized for the one specific location and should know this location best – is often attributed to the fact that, contrary to traditional hydrogeologic or hydrologic models, machine learning models benefit from additional data. However, with the RNDfeat models being as good as the TSfeat and ENVfeat models, we can widely exclude this as a reason in our case (maybe with the exception of the TSfeat model, which seems to perform slightly better than its RNDfeat9 counterpart). Benefiting from additional data or additional
wells, respectively, would presuppose, that the model is able to identify wells that react similar to meteorological inputs based on the static features provided. With the random number features being different for all wells and no meaningful similarity depicted in them, this cannot be the case.

The differences to the single-well models might just as well be attributed to the different meteorological input parameters (rH and Tsin used additionally) and different optimization strategy. Moreover, despite being better on average, we observe a
significant tailing in all global model setups, which the single-well models do not experience to the same degree (figure 3). The tail is the only part of the dataset where single-well models outperform both global models in 18 wells.

**Table 3.** Mean, lower (10%) and upper (90%)-percentile NSE of the 10 ensemble members for all model variants as well as the mean NSE for single-well models as published in Wunsch et al. (2022a).

| variant | NSE ($Q_{10}$) | NSE ($Q_{50}$) | NSE ($Q_{90}$) |
|---|---|---|---|
| single-well | - | **0.8134** | - |
| ENVfeat (IS) | 0.8026 | **0.8213** | 0.8397 |
| RNDfeat18 (IS) | 0.7909 | **0.8215** | 0.8457 |
| TSfeat (IS) | 0.8028 | **0.8229** | 0.8395 |
| RNDfeat9 (IS) | 0.7777 | **0.8135** | 0.8399 |
| DYNonly (IS) | 0.7094 | **0.7347** | 0.7554 |
| ENVfeat (OOS) | 0.3977 | **0.6750** | 0.7685 |
| RNDfeat18 (OOS) | 0.4156 | **0.6767** | 0.7707 |
| TSfeat (OOS) | 0.4590 | **0.6914** | 0.7697 |
| RNDfeat9 (OOS) | 0.4433 | **0.6817** | 0.7710 |
| DYNonly (OOS) | 0.7103 | **0.7326** | 0.7518 |





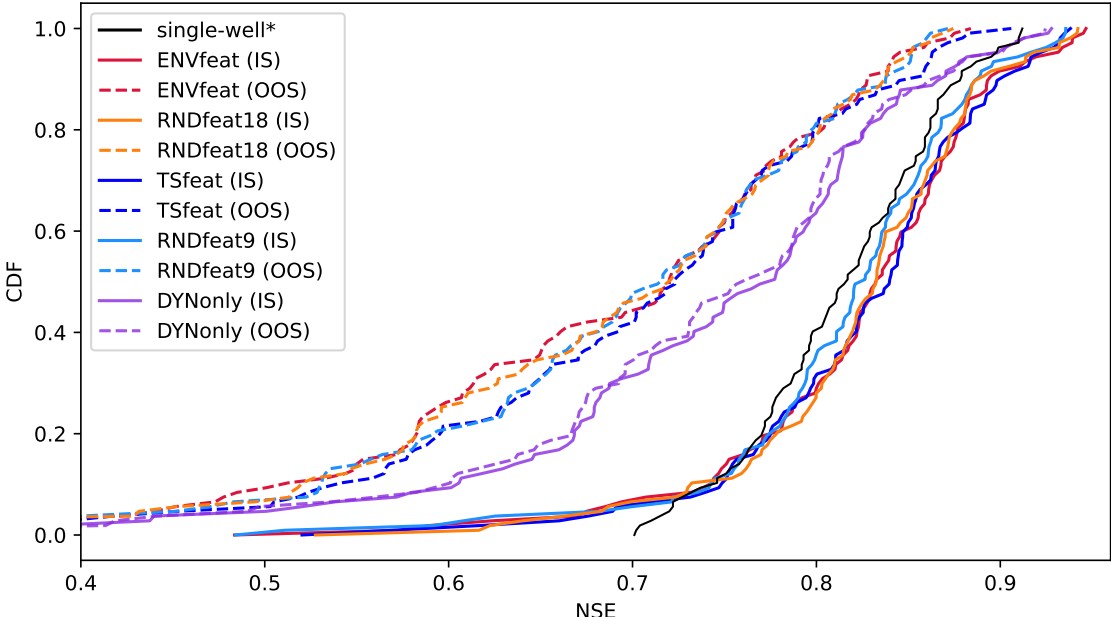

**Figure 3.** Cumulative distributions of NSE for the model variants ENVfeat, TSfeat, RNDfeat9, RNDfeat18 and DYNonly in in-sample mode (IS) and out-of-sample mode (OOS) against the performance of the single-well models by Wunsch et al. (2022a) (*). Lines represent sorted median NSE scores of 10 ensemble members. A version that includes the ensemble ranges as envelopes is shown in Fig. A4.

Figure 3 also shows the out-of-sample (OOS) performance of the global models, represented by 10-fold cross validation runs. As expected, there is a sharp decrease in model performance when a time series' test data is predicted by a model that never saw the time series' training data. However, with a mean NSE of 0.675 (down from 0.8213, see table 3) for the ENVfeat model, and a mean NSE of 0.6914 (down from 0.8229) for the TSfeat model, on average the OOS models perform surprisingly well, especially since some of the performance loss might be attributed to the compromise of 10-fold CV (instead of Leave-One-Out-CV), where about 10% of possible training data is lost compared to the IS setting. These indications of model robustness are counteracted by the large seeding spread associated with both OOS models (fig. 4 or fig A4), and an amplification of the tailing effect. Also, as in the IS case, the RNDfeat models perform nearly identical to the ENVfeat and the TSfeat model (fig. 3), indicating that the model is not able to truly generalize well based on the provided static features, neither on the environmental nor on the time-series features. While the TSfeat model performs at least slightly better than its RNDfeat9 counterpart, which could at least partly support our initial hypothesis that the TSfeat model would outperform the ENVfeat model since static time series features are deemed to be informationally more complete and static environmental features suffer from high uncertainty, the differences are minor and could also be influenced by the relatively low number of 10 ensemble members, showing a large range (fig. 4 or fig A4). In the median, the ranges of NSE values at individual wells for different model seed realizations for the ENVfeat and TSfeat model are around 0.5 in the OOS setting, with minimum range values around 0.08 and maximum range values of more than 1. Even though the spread is one magnitude smaller in the IS setting for all models (medians hovering





around 0.05, see figure 4), this is some significant spread and shows that even if different model runs have the same NSE on the global level, they will have significantly different outcomes on the level of the individual well.

Most strikingly, the DYNonly model, having no static features at all, clearly outperforms all models with static features in the OOS setting (fig. 3, tab. 3). This indicates that the static features even seem to hamper the global entity-aware models to learn a meaningful relationship that is generalizable from the static features in the out-of-sample setting. Furthermore, the OOS performance of the DYNonly model is equally good as its IS performance (down to the third decimal, see table 3), even though it relies on 10 % less training data. This implies information saturation, meaning that all information needed to reach the IS

performance of the DYNonly model can be found in a significantly smaller subset of the dataset.

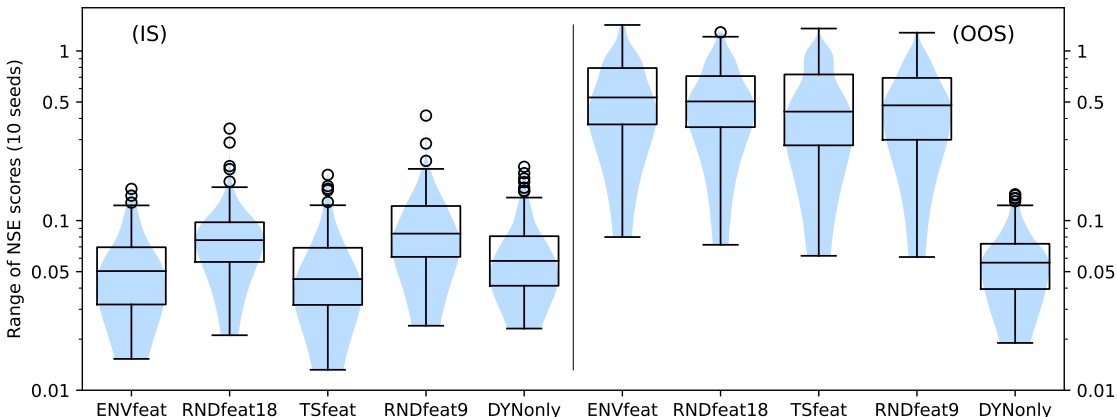

**Figure 4.** Range of NSE scores of the 10 ensemble members of all model variants in in-sample mode (IS) and out-of-sample mode (OOS).

More interesting insights can be drawn on the level of individual well predictions (fig. 5, see supplements for all other wells). Basically, there are two groups of wells, where all wells show more or less the behaviour of one of the two groups, with smooth transitions in between. In the first group, exemplarily shown by well BB_30400591 (fig. 5, top), the predictions in the IS setting of the ENVfeat, TSfeat, and RNDfeat models match rather well the observations with NSE for all models above 0.8, while

the DYNonly model clearly fails (NSE = 0.169). Moreover, the predictions of the ENVfeat, TSfeat, and RNDfeat models are quite similar, confirming their overall similar performance. Thus, the models seem to obtain important information from the static features, that allow to better train the model to the individual behaviour of the well, while (as already postulated above) the kind of static information, environmental, time-series or random, doesn't seem to matter. On the other hand, the DYNonly model obviously lacks information about the special behaviour of the well, and probably predicts some average reactions to

the inputs, that do not work well for the considered well. In the OOS setting, the predictions of all models are quite similar, or in other words, the predictions of the ENVfeat, TSfeat and RNDfeat approach the one of DYNonly model, which is nearly the same as in in-sample setting. The ENVfeat, TSfeat and RNDfeat have obviously lost their ability to predict the individual behaviour of the considered well, as the well was not included into training, and the model was obviously not able to generalize





the relevant information from other wells' static data. Thus, there is a drastic drop in model performance between IS and OOS

setting. Other good representatives for this group are e.g. well BW_107-666-2, and well SH_10L53126001 (see supplements).

In the second group, represented by well SH_10L62060004 (fig. 5, botton), all model predictions in the IS and OOS setting, including the DYNonly model, are quite similar. All models perform well with NSE > 0.8 and there is no obvious performance drop between IS ans OOS. Our interpretation is, that these are wells show a more "average" behaviour in terms of their reaction to the meteorological inputs, i.e. the "average" reaction to the meteorological inputs that is learned by the DYNonly

model. Therefore, the additional information provided by the static parameters does not improve model performance in IS setting. Conversely, it also doesn't negatively influence model performance when this information is missing OOS, explaining the absent or minor drop in performance. Other good representatives for this group are e.g. well BW_124-068-9, and well NI_200001722.

### 4.2 Permutation Feature Importance

By looking at the models' input feature importance, further insights can be gained. We applied Permutation Feature Importance to detect the relative importance of individual input features in the trained models. Figure 6 shows the feature importance of the ENVfeat, TSfeat and RNDfeat models, separately for static and dynamic features (dynamic feature importance includes all model variants).

The first thing to note is that every individual dynamic feature is much more important than any of the static features,

as the permuted MSE increase is higher by orders of magnitude for dynamic features. Thereby, as expected, precipitation and temperature are the most important features, followed by the sinus curve fitted to temperature. Relative humidity is least important. Even though there is some instability involved in these results, with especially Tsin and rH, but also P and T experiencing heavy outliers up to 2 magnitudes above the median, by and large this confirms the findings of previous studies (Wunsch et al., 2021b).

Among static features, we find much more indifference between individual features (figure 6). Among static environmental features, CLC land cover comes out on top. This seems plausible because of its conceptual importance, and because it is the only feature representing land cover (although comprising 11 categories), unifying all information of land cover forcings, i.e. being informationally dense for the model. However, all environmental features show about the same feature importance as their random counterparts, confirming conclusions drawn in section 4.1 that they do not contribute any meaningful and thus

generalizable information to the model.

Static time series features are a bit more sensitive, with high pulse duration (HPD) and annual periodicity (P52) outpacing all other time series features by some margin. But feature importance remains on a very low level for all other time series features, which are even surpassed by the average (relative) importance of the nine random features, the exception being high pulse duration (HPD). This allows the conclusion that this is the only static feature that might provide some meaningful/generalizable

information to the model. It also could be the reason, that the TSfeat model at least slightly outperforms the RNDfeat9 model (compare fig. 3).







**Figure 5.** Examples of predictions of groundwater levels in the test period (2012-2015) for two wells, representing groups of similar behaviour (see text for details).

In general, however, the most important finding to take from this result is the fact that all static features are orders of magnitude less important than the dynamic features, which implies that the model draws the majority of information used for prediction from the shared dynamic features. This can be used as an explanation for the finding that ENVfeat, TSfeat and

RNDfeat models perform almost identically (see fig 3), while DYNonly is able to outperform them out-of-sample.





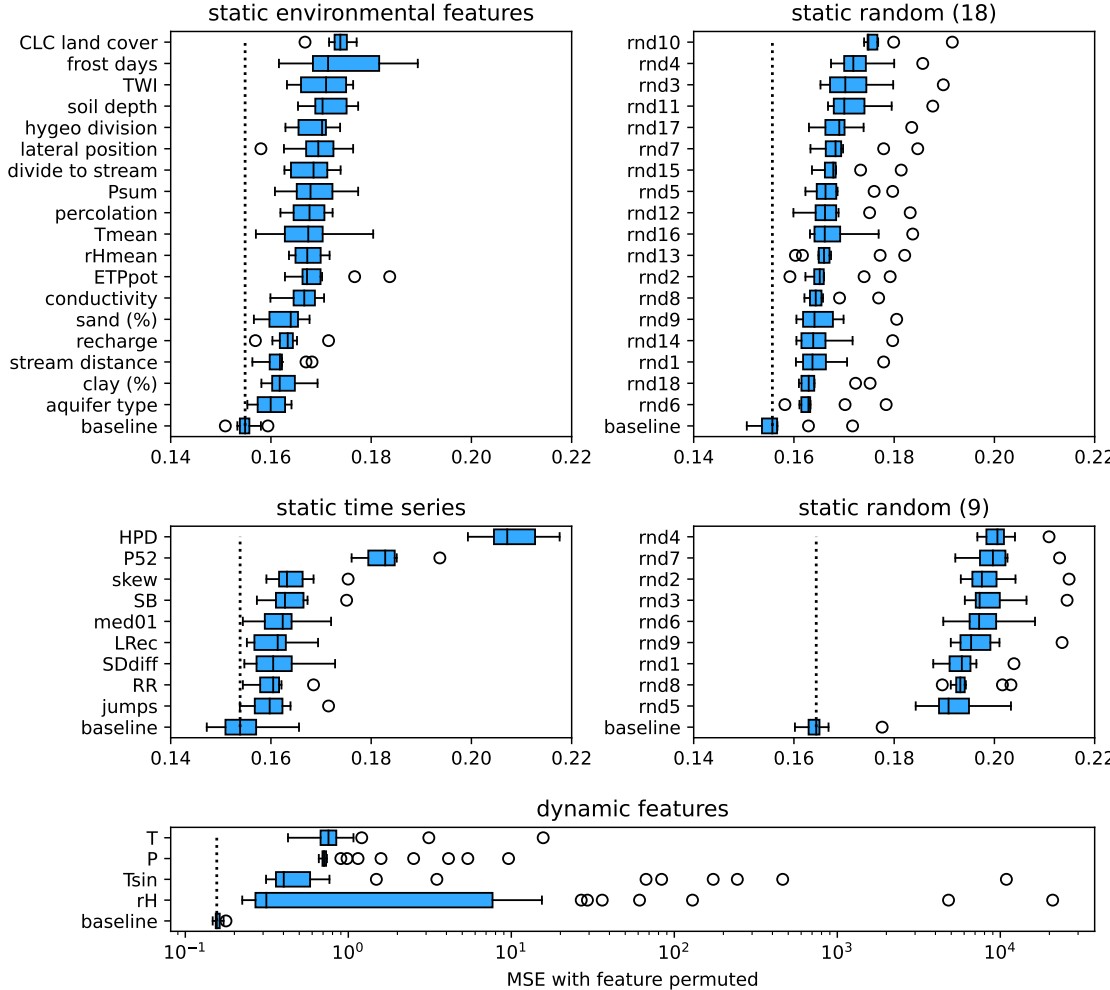

**Figure 6.** Permutation feature importance of the ENVfeat, TSfeat and RNDfeat models, separately for static and dynamic features (dynamic feature importance includes all model variants).

## 5 Conclusions

The results of our work allow two main conclusions. First, in the in-sample setting, entity-aware global models work well and their performance can keep up with those of single-well models. All proposed model variants reached slightly better scores than the state-of-the-art single station model. However, contrary to our initial hypothesis, the TSfeat model does not outperform the

ENVfeat model. Moreover, the RNDfeat models - having random integers instead of "meaningful" static features - performed equally well as the TSfeat and ENVfeat variants. Against the backdrop of the in-sample DYNonly model (trained without static features, only on meteorologic input), which had severely reduced performance, it is evident that this is because all tested sets of static features appear to only act as unique identifiers, enabling the model to differentiate between time series and memorizing



their unique behaviour, but not to establish meaningful system-characterizing relationships based on the static features. Thus,
we conclude that the models do not learn adequately from wells with similar information as provided by the static features.
It may not be worth the effort to gather supposedly meaningful data, since random numbers might work just as well (as long
as a decent number of random features is provided). This finding is in accordance with studies that have been carried out in
rainfall-runoff modelling (Li et al., 2022b). Also, observed performance improvement over single-well models might just as
well be due to architectural differences and the incorporation of additional dynamic input features (namely rH and Tsin) that
weren't considered in the published single-well model results used as comparison. In other words, the models introduced here
perform better, but not necessarily for the reason of being global or entity-aware, according to the commonly made claim that
global models profit from additional similar data.

Second, out-of-sample performance of all model variants with static features expectedly decreases significantly. In general
this is still a respectable performance, making a case for good generalizability in principle. However, the DYNonly model
significantly outperformes TSfeat, ENVfeat and RNDfeat variants in this setting. This makes evident that static features, acting
as unique identifiers in-sample, obscure learning of the only true meaningful or causal connection, namely of dynamic (mete-
orologic) input features to groundwater levels out-of-sample (i.e. when not included in training). In other words, the models
are not able to learn that wells with a similar static feature combination should react similar to meteorological dynamic fea-
ture inputs in terms of groundwater level output. Instead, model skill is almost entirely based on learning from the dynamic
input features. This might not come as a surprise for the environmental features, which were deemed to be afflicted with high
uncertainties, but for the time-series features, since these proved in previous works to be well-suited to describe groundwater
dynamics as a result of its reaction to meteorological inputs. Thus, our results suggest only a temporal generalizability potential
- however valuable in itself - of entity-aware models, but lack evidence of true spatial generalizability potential - which remains
the overarching aim of the field. However this stands on the presented database, which, as stated above, might be too small, not
diverse enough, and/or biased.

The tasks set by these conclusions are clear. First, since the dataset might just not contain enough data to take full advantage
of global models, we plan to investigate this with a larger dataset, that covers groups of wells with several similarities as well as
dissimilarities in a future study. The hypothesis is, that when more wells with similar meaningful static information are included
in the dataset, the entity-aware model might then be able to better learn and generalize from the provided features. Second,
our study revealed the glaring research gap of finding and compiling meaningful environmental descriptors of groundwater
dynamics with true predictive power. The hydrogeologic discipline lacks large scale datasets of the kind. This severely restricts
the development of hydrogeology as a machine learning research field, and the establishment of neural network models with
physiographically meaningful internal structures, as was pursued in this study.

*Code and data availability.* The original groundwater level data are available free of charge from the respective local authorities: LUBW
Baden-Wuerttemberg, LfU Bavaria, LfU Brandenburg, HLNUG Hesse, LUNG Mecklenburg-Western Pomerania, NLWKN Lower-Saxony,
LANUV North Rhine-Westphalia, LfU Rhineland-Palatinate, LfULG Saxony, LHW Saxony-Anhalt and LLUR Schleswig-Holstein. With



the kind permission of these local authorities, the processed groundwater level data have been published by Wunsch et al. (2021a). Mete-
orological input data was derived from the HYRAS dataset (Rauthe et al., 2013; Frick et al., 2014), which can be obtained free of charge
for non-commercial purposes on request from the German Meteorological Service (DWD). The Python code files are available on GitHub:
400 https://github.com/KITHydrogeology/2023-global-model-germany.





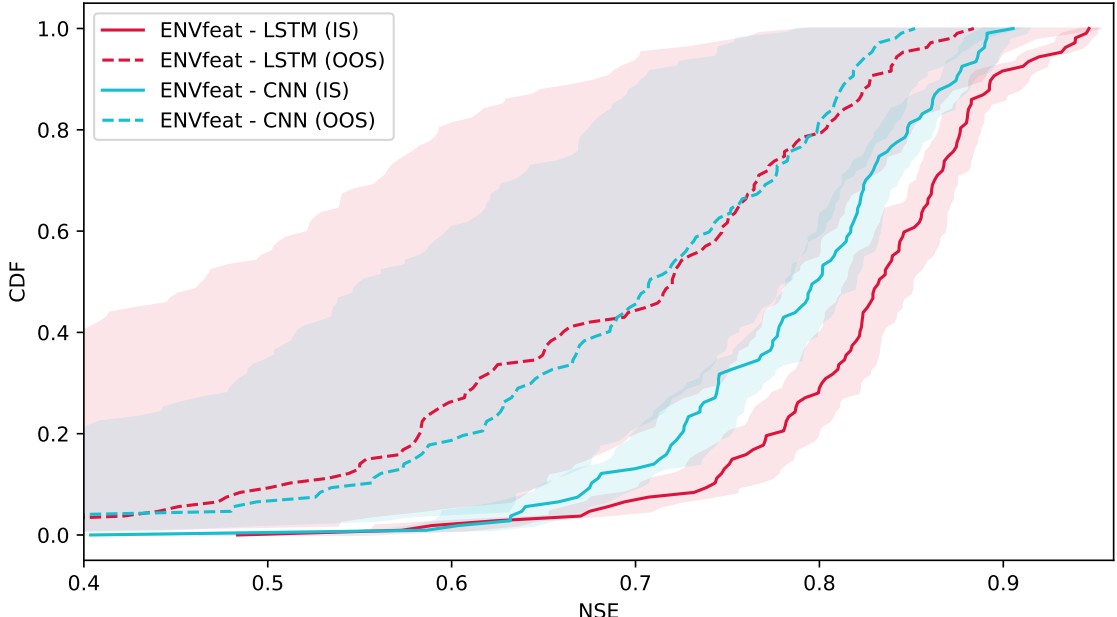

**Figure A1.** Comparing the performance of LSTM to CNN on the basis of the ENVfeat model variant. For the CNN model, the LSTM layer in the dynamic model thread is replaced with a CNN layer (followed by batchnorm and maxpool1D). The figure shows the models' performance in an in-sample (IS) and out-of-sample (OOS) setup. While CNNs and LSTMs perform almost equivalent in the OOS mode, CNNs are clearly inferior to LSTMs in the IS mode. Thus, LSTMs were used in this study.





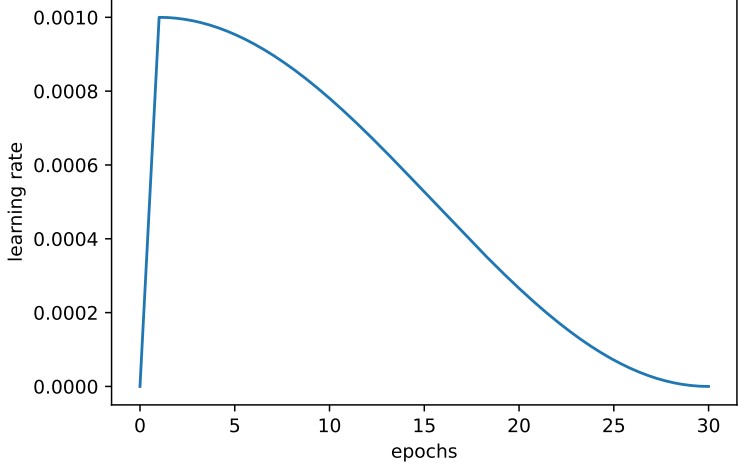

**Figure A2.** Visualisation of the learning rate schedule used in this study. It consists of one warmup epoch where the learning rate linearly increases from 0 to 0.001, followed by 29 epochs of cosine-shaped learning rate decline.





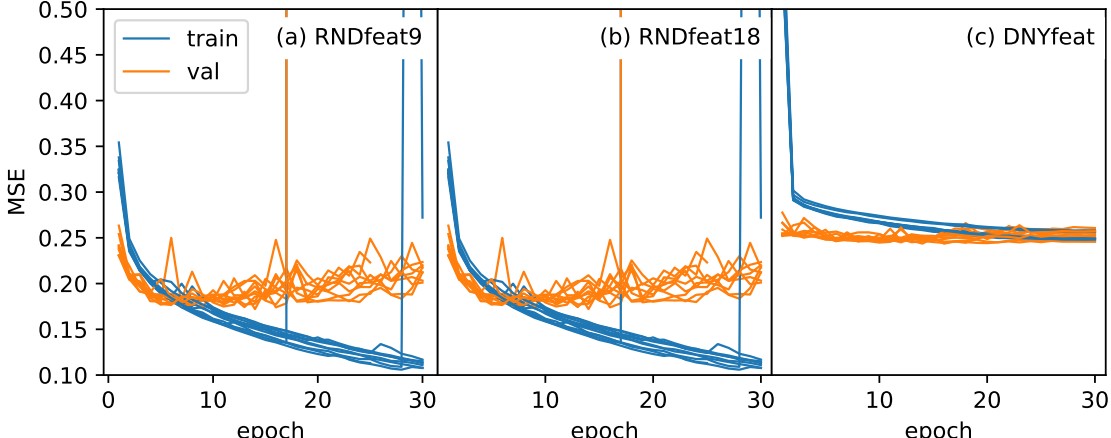

**Figure A3.** Losscurves of the training and validation period for the 10 different seed initialization runs for the (a) RNDfeat9 model, (b) RNDfeat18 model and (c) DYNonly model.



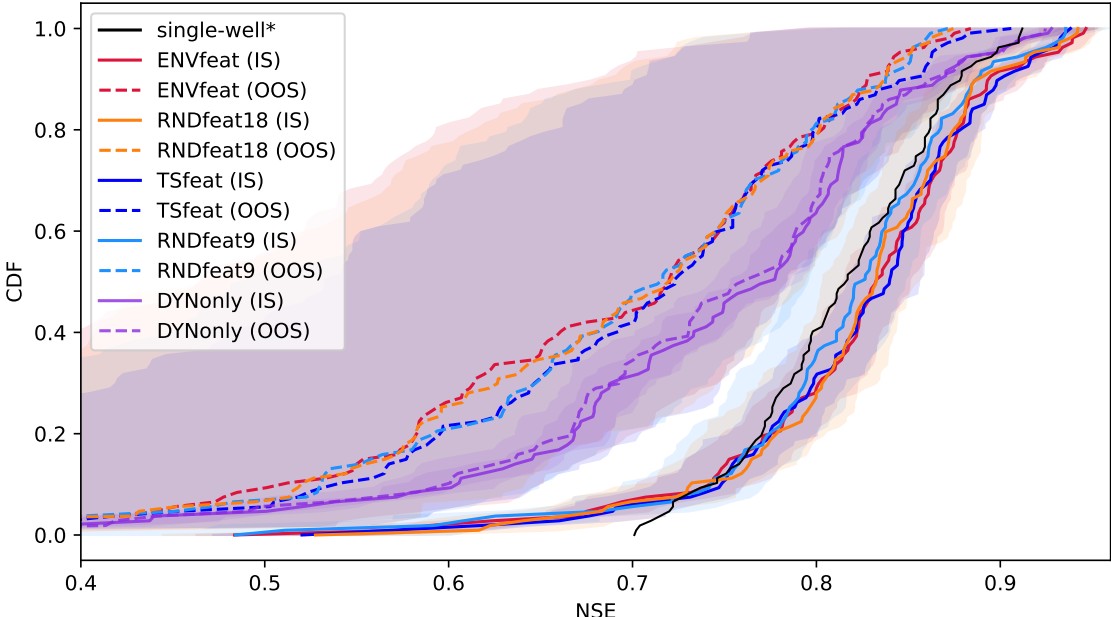

**Figure A4.** Cumulative distribution function of NSE of the model variants ENVfeat, TSfeat, RNDfeat and DYNonly in in-sample mode (IS) and out-of-sample mode (OOS) against the performance of the single-well models by Wunsch et al. (2022a) (*). Lines represent sorted median NSE scores of 10 ensemble members, envelopes represent ranges of the ensemble forecasts excluding the worst and best member.



*Author contributions.* BH and TL conceptualized the study, wrote the code, validated and visualized the results, and wrote the original paper draft. All three authors contributed to the methodology and performed review and editing tasks. TL supervised the work.

*Competing interests.* The authors declare that they have no conflict of interest.

*Acknowledgements.* All programming was done in Python version 3.9 (Van Rossum and Drake Jr, 1995) and the associated libraries, in-
405   cluding NumPy (Harris et al., 2020), Pandas (McKinney et al., 2010), Tensorflow (Abadi et al., 2016), Keras (Chollet et al., 2015), SciPy (Virtanen et al., 2020), Scikit-learn (Pedregosa et al., 2011) and Matplotlib (Hunter, 2007).



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
