# Peer review of "On the challenges of global entity-aware deep learning models for groundwater level prediction"

_Hydrology and Earth System Sciences, 2023_

## Referee Comment (RC2)

Manuscript title: On the challenges of global entity-aware deep learning models for groundwater level prediction

**Overview**

The authors develop an entity-aware deep learning model for spatially and temporally continuous groundwater level prediction using a combined Long Short-Term Memory (LSTM) and Multi-Layer Perceptron (MLP) network. They rely on ground observations from 108 wells in Germany and other dynamic and static predictor data obtained from multiple sources to train, validate, and test the model. The authors also perform some interesting comparisons of four model variants, namely, the time series feature-driven model (TSFeat), environmental feature-driven model (ENVfeat), random static features (RNDfeat), and dynamic inputs only features (DYNonlyfeat). While there are some issues with spatial generalizability in the out-of-sample setting, the model shows satisfactory performance with the Nash-Sutcliffe Efficiency (NSE) > 0.8 in an in-sample setting.

Overall, the manuscript is generally well-structured, with detailed explanations of the methodologies and data. However, the following comments should be addressed before this manuscript is published.

**Major Comments**

- The authors should discuss relevant literature that incorporates process-based, machine-learning, or hybrid models and remote sensing data for groundwater level monitoring. The Introduction section should highlight the relevance of this topic more and refer to some of the negative impacts of groundwater depletion and why groundwater level monitoring is essential.

Hasan, M.F., Smith, R., Vajedian, S. et al. Global land subsidence mapping reveals widespread loss of aquifer storage capacity. *Nat Commun* 14, 6180 (2023). https://doi.org/10.1038/s41467-023-41933-z

Herrera-García, G. et al. Mapping the global threat of land subsidence. Science 371, 34–36 (2021). https://doi.org/10.1126/science.abb8549

Famiglietti, J. The global groundwater crisis. Nature Clim Change 4, 945–948 (2014). https://doi.org/10.1038/nclimate2425

Wada, Y. et al. Global depletion of groundwater resources. Geophys. Res. Lett. 37, 1–5 (2010). https://doi.org/10.1029/2010GL044571

Faunt, C.C., ed., 2009, Groundwater Availability of the Central Valley Aquifer, California: U.S. Geological Survey Professional Paper 1766, 225 p.

Li, B., et al. (2019). Global GRACE Data Assimilation for Groundwater and Drought Monitoring: Advances and Challenges. Water Resources Research, 55(9), 7564–7586. https://doi.org/10.1029/2018WR024618

Ahamed, A., Knight, R., Alam, S., Pauloo, R., & Melton, F. (2022). Assessing the utility of remote sensing data to accurately estimate changes in groundwater storage. *Science of The Total Environment*, *807*, 150635. https://doi.org/10.1016/j.scitotenv.2021.150635

Bierkens, M. F. P., & Wada, Y. (2019). Non-renewable groundwater use and groundwater depletion: a review. *Environmental Research Letters*, *14*(6), 063002. https://doi.org/10.1088/1748-9326/ab1a5f

• In addition to the predictor data summary, the authors should include a description of the predictor data listed in Tables 1 and 2 and the associated uncertainty. In Table 2, what does self-derived mean? Would snow water equivalent and soil moisture be helpful as additional predictors to capture the groundwater dynamics better? The authors should make a stronger case for selecting HYRAS 3.0 than other globally available land-surface models like the Global Land Data Assimilation System (GLDAS), which provides spatially and temporally continuous estimates of various hydrological processes acting as critical drivers of groundwater dynamics.

Rodell, M., et al. (2004). The Global Land Data Assimilation System, Bull. Amer. Meteor. Soc., 85(3), 381-394. https://doi.org/10.1175/BAMS-85-3-381

Razafimaharo, C., Krähenmann, S., Höpp, S. et al. New high-resolution gridded dataset of daily mean, minimum, and maximum temperature and relative humidity for Central Europe (HYRAS). Theor Appl Climatol 142, 1531–1553 (2020). https://doi.org/10.1007/s00704-020-03388-w

• The authors should include the model forecasts beyond January 2016. While it may be challenging to obtain in-situ groundwater levels between 2016-present, it would be interesting to observe how the model predictions compare to the GRACE- and GRACE Follow-On (GRACE-FO)-based total water storage changes (https://grace.jpl.nasa.gov/data/data-analysis-tool/) at a regional or national scale. This comparison would serve as an additional model validation and strengthen the manuscript.

• There should be an additional section (or a subsection within the Introduction) describing the study area and related studies on groundwater level changes. Also, the spatial distribution of the 108 well locations should be shown on a map.

• What are the 11 land cover classes in the CLC data? How are these used in the model? Can the categories be reduced by aggregating to a base class? E.g., crops aggregated to 'Agriculture,' urban/industry to 'Urban,' and so on? Is there no significant change in built-up or irrigated areas within the temporal domain of the model? The potential effects of land use changes on the model performance should be discussed. Also, the percentage of land use classes

should be described in the Study Area section.

- The corresponding time series of the dynamic predictors for the two wells in Figure 5 should be added and tied up with the discussion related to the permutation feature importance.

- Evapotranspiration (ET) is the second largest component of the water cycle after precipitation (https://openetdata.org/what-is-evapotranspiration) and is a critical driver of groundwater use, which, in turn, is correlated to groundwater levels (Majumdar et al., 2020; 2022; Brookfield et al., 2023; Melton et al., 2021; Senay et al., 2022). Why didn't the authors include it as a dynamic predictor and instead rely on the potential ET (Table 2)? While the OpenET and the Landsat-derived actual ET products are currently available only over the conterminous United States (CONUS), the globally available 500 m MOD16 actual ET is available within the temporal domain of the model. Thus, the authors should justify the choice of their predictors.

Majumdar, S., Smith, R., Butler, J. J., & Lakshmi, V. (2020). Groundwater withdrawal prediction using integrated multitemporal remote sensing data sets and machine learning. *Water Resources Research*, 56(11), e2020WR028059. https://doi.org/10.1029/2020WR028059

Majumdar, S., Smith, R., Conway, B. D., & Lakshmi, V. (2022). Advancing remote sensing and machine learning-driven frameworks for groundwater withdrawal estimation in Arizona: Linking land subsidence to groundwater withdrawals. *Hydrological Processes*, *36*(11), e14757. https://doi.org/10.1002/hyp.14757

Brookfield, A. E., Zipper, S., Kendall, A. D., Ajami, H., & Deines, J. M. (2023). Estimating Groundwater Pumping for Irrigation: A Method Comparison. *Groundwater*. https://doi.org/10.1111/gwat.13336

Melton, F., et al. (2021). OpenET: Filling a Critical Data Gap in Water Management for the Western United States. *JAWRA Journal of the American Water Resources Association*. https://doi.org/10.1111/1752-1688.12956

Senay, G. B., et al. (2022). Mapping actual evapotranspiration using Landsat for the conterminous United States: Google Earth Engine implementation and assessment of the SSEBop model. *Remote Sensing of Environment*, 275, 113011. https://doi.org/10.1016/j.rse.2022.113011

Running, S., Mu, Q., Zhao, M. (2021). MODIS/Terra Net Evapotranspiration 8-Day L4 Global 500m SIN Grid V061 [Data set]. NASA EOSDIS Land Processes Distributed Active Archive Center. Accessed 2023-10-17 from https://doi.org/10.5067/MODIS/MOD16A2.061.

- Lines 50-60: While the proposed machine learning-based method of using multiple wells to develop an entity-aware global groundwater level prediction model is new, earlier studies have integrated remote sensing, climate, and hydrogeologic data in a machine learning framework for estimating annual groundwater withdrawals (Majumdar et al., 2020; 2021; 2022; Wei et al., 2022) and land subsidence (Smith & Majumdar, 2020; Hasan et al., 2023). For the studies on groundwater withdrawal estimation, a single machine learning model was trained and validated using in-situ pumping measurements from multiple wells across vast geographical areas (states of Kansas and Arizona in the U.S.). Thus, the authors should clearly convey that the novelty lies in groundwater level monitoring rather than the entire hydrogeology domain.

Majumdar, S., Smith, R., Butler, J. J., & Lakshmi, V. (2020). Groundwater withdrawal prediction using integrated multitemporal remote sensing data sets and machine learning. *Water Resources Research*, 56(11), e2020WR028059. https://doi.org/10.1029/2020WR028059

Majumdar, S., Smith, R., Conway, B. D., Butler, J. J., Lakshmi, V., & Dagli, C. H. (2021). Estimating Local-Scale Groundwater Withdrawals Using Integrated Remote Sensing Products and Deep Learning. *2021 IEEE International Geoscience and Remote Sensing Symposium IGARSS*, 4304–4307. https://doi.org/10.1109/IGARSS47720.2021.9554784

Majumdar, S., Smith, R., Conway, B. D., & Lakshmi, V. (2022). Advancing remote sensing and machine learning-driven frameworks for groundwater withdrawal estimation in Arizona: Linking land subsidence to groundwater withdrawals. *Hydrological Processes*, *36*(11), e14757. https://doi.org/10.1002/hyp.14757

Wei, S., Xu, T., Niu, G.-Y., & Zeng, R. (2022). Estimating Irrigation Water Consumption Using Machine Learning and Remote Sensing Data in Kansas High Plains. *Remote Sensing*, *14*(13), 3004. https://doi.org/10.3390/rs14133004

Smith, R., & Majumdar, S. (2020). Groundwater storage loss associated with land subsidence in Western United States mapped using machine learning. *Water Resources Research*, *56*(7), e2019WR026621. https://doi.org/10.1029/2019WR026621

Hasan, M.F., Smith, R., Vajedian, S. et al. Global land subsidence mapping reveals widespread loss of aquifer storage capacity. *Nat Commun* 14, 6180 (2023). https://doi.org/10.1038/s41467-023-41933-z

**Minor Comments**
➢ Line 94- Fix typo: followed by a brief *introduction.*
➢ What are the spatial resolutions of the predictor data listed in Table 2? How do the authors map the groundwater wells to these gridded raster datasets?
➢ Report other error metrics like the coefficient of determination ($R^2$), root mean square error (RMSE), and the mean absolute error (MAE) in Table 3.
➢ The CLC acronym is not defined.

➢ Do the authors scale all the features? What scaling is applied?

➢ Lines 235-240: For the out-of-sample setting, are the scores only calculated for the testing period of a well that has been left out of model training? Why not calculate the score for the entire period?

---

## Author Comment (AC1)

The authors have presented an application of machine learning techniques to create a global model of groundwater levels in Germany. They compared two different input model settings: one with static features and one with dynamic features. Additionally, they compared these settings with two reference cases, one with random entity variables and a second without entity variables. Their results indicate that both model settings perform well under in-sample conditions, but their performance diminishes in out-of-sample conditions. The work aligns with the growing trend in this field, introducing entity-aware methods to hydrogeology and yielding promising results. I believe the paper has the potential for publication, but there are some points that need to be addressed before publication.

> We thank Reviewer 2 for his positive assessment. We will address his valuable suggestions as itemized below in our responses (blue).

Limited Machine Learning Methods Tested: The authors only tested two machine learning methods, LSTM and CNN. There are powerful alternatives like Transformers, which have outperformed LSTM in other studies (https://doi.org/10.1016/j.apr.2023.101833). LSTM models are designed for handling transient conditions, whereas CNNs are adapted to do so. There are various other methods like extreme gradient boosting that have been applied in hydrogeology that can be a powerful alternative to CNN (cite: https://doi.org/10.1016/j.watres.2023.119745, https://doi.org/10.1007/s10661-020-08695-3, https://doi.org/10.1016/j.scitotenv.2021.151065). The authors should explain why they chose not to explore more advanced models before attributing out-of-sampling prediction issues to dataset limitations.

> Reviewer 1 is right in that potentially more powerful AI models might be available, when considering absolute performance. However, there are sound reasons why we chose to stick with LSTM (and CNN) models. First of all, as outlined in the introduction, and as elaborated in detail in a recent review by Tao *et al.* (2022) cited in the paper, LSTM and CNN are currently the predominant model class in groundwater level modelling. This is because they consistently deliver high-class performance while maintaining some degree of model simplicity, thereby satisfying Occam's Razor. The mentioned extreme gradient boosting proved to be powerful in groundwater quality modeling. However, this is a related but different field from groundwater level modeling with a number of significant differences, making methods not directly transferable. Transformers seemed promising 1 or 2 years ago, and their suitability as a general-purpose method beyond its original domain (language modeling) are increasingly called into question because they can be outperformed by more simple linear DL models (Zeng *et al.*, 2023), notably across the board (Das *et al.*, 2023). In conclusion, while testing different newer/other models to increase overall performance is desirable, this needs a separate study with careful model selection and a clear experimental design geared towards appropriate research questions. This was out of scope for the study at hand, which deals with theoretical questions regarding entity awareness. Regarding this manuscript, we will further elaborate the choice of LSTM and CNN over other methods in the papers' introduction along the line of above argumentation.

Data Fusion Method Comparison: In Line 164, the authors propose a new data fusion method. Did they compare this method to existing methods to demonstrate its benefits and limitations? Comparing it to cited methods would provide valuable insights.

> The data fusion method used in the study is not new but well-established in various fields (e.g. Liu et al. 2022, Miebs et al., 2020). However it is true that it in other studies in neighboring disciplines (e.g. Kratzert et al. 2019), often a more simplistic approach of duplicating the static input features in each time step is used. As discussed e.g. in Miebs et al.

(2020), this is not an optimal choice for an RNN architecture. Such an approach leads to a significant increase in the number of RNN parameters, since duplicated static features are evaluated each time for every sequence. Moreover, these duplicate data do not add any meaningful additional information. As a consequence, the training of such a network is both memory- and time-consuming (in our case it tripled the computation time in some initial experiments and, moreover, showed rather similar results). We are also aware that there are more sophisticated approaches of combining dynamic and static features in machine learning models (like using static features to initialize cell states in LSTMs or to learn attention weights). But even though we agree with the reviewer, that a comparison of different data fusion methods would provide valuable insights, this was not the scope of our study. Again, we chose a well-established method that yielded good results as a basis for our actual question of research.

Introduction: The introduction lacks a description of the study area, which should be addressed since the model applies to a single case study. Adding a figure depicting the study area and well distribution would enhance the paper's context.

We thank Reviewer 1 for the useful suggestion. We will add a map with study are to the revised manuscript.

Well data selection (Line 104): Quantitatively explain what "spatial coverage as representative as possible" means. Clarify if there is a minimum distance between wells, data density per area, or any specific criteria used for well selection. Provide the original dataset size from which data was picked.

The groundwater data used in this study was primarily chosen because it is a readily available dataset that is already published (Wunsch et al., 2022), enabling reproducibility and circumventing the need to assemble and publish a new dataset, which is a very time consuming and painful process in Germany due to data accessibility problems, unnecessarily delaying paper publication in a competitive field. We will remove the note on data distribution and will instead expand the justification to use a pre-published dataset for reproducibility and refer the reader to the published paper of Wunsch et al. (2022). We hope this, together with the visual evidence of the datasets' actual distribution from the map (see comment above) will suffice Reviewer 1.

Upscaling (Line 136): Elaborate on the importance of not having too fine-grained categorical data. Describe the upscaling process and how the authors ensured that each category is correctly represented. Provide references or explore the effect of upscaling on training and prediction.

We realize the passage in the manuscript is written ambiguously. There was no upscaling involved on our side. Instead, we simple had several categorical datasets of the same type at hand and chose the one with less categories. As an example, for soil type, we had the choice between a product called "buek200" with a scale of 1:200,000 which has more than 550 different soil type categories, and another product called "buek5000" with a scale of 1:5,000,000 which has only 23 categories, and which is a generalized version of buek200 on a larger scale. Buek5000 was chosen because communality of soil type classes between groundwater locations would be impossible with buek200: Using buek200 would almost certainly lead to every location having a unique soil type, thus not allowing any study of entity awareness. This is the selection process we wanted to describe in line 136. We will elaborate this better in the text.

MLP Classifier (Line 168): Explain the advantages of adding an MLP classifier rather than providing static features directly. Address concerns about uncertainty propagation due to MLP output in the concatenation.

> We are not sure what reviewer #1 means here. We did not use an MLP classifier, but an MLP for processing the static features (as a regression, not classification). We refer to the point above on data fusion. The chosen approach is a well-established method for the incorporation of static features into recurrent neural networks, see e.g. Miebs et al. 2020.

MLP Output Nodes (Line 170): Specify the number of output nodes in the MLP.

> All numbers of output nodes are given in the text. In line 170 we write "with one fully connected (Dense) layer of size 128 in the static model thread."

Validation MSE (Figure 2): Explain the phenomenon where the validation MSE is smaller than the training MSE, especially in the initial epochs. This could indicate a bias in the validation dataset, and clarification is needed.

> We thank Reviewer 1 to point us towards this aspect that is not sufficiently elaborated yet. The shape of the losscurves indeed indicates bias in validation data, which is probably also the case for the test data. This is due to the fact that validation data is not uniformly sampled over the whole time period, but the fixed time period of 2008-2011 (and test data being 2012-2015), as specified in the paper. These periods are the most recent data period, and it was consciously chosen as fixed due to the fact that the aim is forward prediction of groundwater levels, meaning that the most recent groundwater levels are most representative for a possible future (as opposed of choosing rolling time periods for validation and testing). We will elaborate this better in the revised version of the manuscript.

Feature Importance (Line 245): Suggest using the SHAP method (Lundeberg and Lee, 2017, https://doi.org/10.48550/arXiv.1705.07874) for more stable feature ranking, as it has been used effectively in similar studies (Ransom et al., 2020, https://doi.org/10.1016/j.scitotenv.2021.151065).

> Thank you for this suggestion. We are aware of the SHAP method and we have also used it ourselves before in other studies (i.e. Wunsch et al. 2022). However, it is quite computationally intensive, and that is why we preferred Permutation Feature Importance here. We will better explain the choice of Permutation Feature Importance over other XAI methods (like SHAP or Layerwise Relevance Propagation) in the revised version and we will take the suggestion up in the manuscripts' discussion as a potential alternative method.

**Literature**

Das, A. *et al.* (2023) 'Long-term Forecasting with TiDE: Time-series Dense Encoder'. arXiv. Available at: http://arxiv.org/abs/2304.08424 (Accessed: 13 November 2023).

Kratzert, F. *et al.* (2019) 'Towards learning universal, regional, and local hydrological behaviors via machine learning applied to large-sample datasets', *Hydrology and Earth System Sciences*, 23(12), pp. 5089–5110. Available at: https://doi.org/10.5194/hess-23-5089-2019.

Liu, Q., Yang, M., Mohammadi, K., Song, D., Bi, J., & Wang, G. (2022). Machine Learning Crop Yield Models Based on Meteorological Features and Comparison with a Process-Based Model. Artificial Intelligence for the Earth Systems, 1(4), e220002.

Miebs, G., Mochol-Grzelak, M., Karaszewski, A., & Bachorz, R. A. (2020). Efficient strategies of static features incorporation into the recurrent neural network. Neural Processing Letters, 51(3), 2301-2316.

Tao, H. *et al.* (2022) 'Groundwater level prediction using machine learning models: A comprehensive review', *Neurocomputing*, 489, pp. 271–308. Available at: https://doi.org/10.1016/j.neucom.2022.03.014.

Wunsch, A., Liesch, T. and Broda, S. (2022) 'Deep learning shows declining groundwater levels in Germany until 2100 due to climate change', *Nature Communications*, 13(1), p. 1221. Available at: https://doi.org/10.1038/s41467-022-28770-2.

Zeng, A. *et al.* (2023) 'Are Transformers Effective for Time Series Forecasting?', *Proceedings of the AAAI Conference on Artificial Intelligence*, 37(9), pp. 11121–11128. Available at: https://doi.org/10.1609/aaai.v37i9.26317.

---

## Author Comment (AC2)

**Overview**

The authors develop an entity-aware deep learning model for spatially and temporally continuous groundwater level prediction using a combined Long Short-Term Memory (LSTM) and MultiLayer Perceptron (MLP) network. They rely on ground observations from 108 wells in Germany and other dynamic and static predictor data obtained from multiple sources to train, validate, and test the model. The authors also perform some interesting comparisons of four model variants, namely, the time series feature-driven model (TSFeat), environmental feature-driven model (ENVfeat), random static features (RNDfeat), and dynamic inputs only features (DYNonlyfeat). While there are some issues with spatial generalizability in the out-of-sample setting, the model shows satisfactory performance with the Nash-Sutcliffe Efficiency (NSE) > 0.8 in an in-sample setting.

Overall, the manuscript is generally well-structured, with detailed explanations of the methodologies and data. However, the following comments should be addressed before this manuscript is published.

> We thank Reviewer 2 for his positive assessment. We will address his valuable suggestions as itemized below in our responses (blue). For the sake of a clear overview, all references initially provided by Reviewer 2 were omitted from this answer. They can still be found in the original Review.

**Major Comments**

The authors should discuss relevant literature that incorporates process-based, machine learning, or hybrid models and remote sensing data for groundwater level monitoring. The Introduction section should highlight the relevance of this topic more and refer to some of the negative impacts of groundwater depletion and why groundwater level monitoring is essential.

> Reviewer 2 is right in that the introduction could use some further expansion on the need for groundwater level monitoring and modeling due to threats to sustainable groundwater use as lined out beautifully by the literature provided by Reviewer 2. We will add some more sentences on these aspects in a revised version.
>
> Regarding discussion on process-based models, we refer to line 21-26, where it is argued that numerical models are not suitable for the national scale. Accordingly, there is no national numerical groundwater model that could be discussed. We will add this fact to the discussion in line 21-26, and will expand on alternative applications like e.g. the process-based estimation methods in the central valley study provided by Reviewer 2 – which are, however, also unavailable in Germany.
>
> Regarding remote sensing data studies, we point out that our study focusses primarily on analyzing a general theoretical problem with entity aware models, leaving the question open how remote sensing data could be generally beneficial in this.

In addition to the predictor data summary, the authors should include a description of the predictor data listed in Tables 1 and 2 and the associated uncertainty. In Table 2, what does selfderived mean? Would snow water equivalent and soil moisture be helpful as additional predictors to capture the groundwater dynamics better? The authors should make a stronger case for selecting HYRAS 3.0 than other globally available land-surface models like the Global Land Data Assimilation System (GLDAS), which provides spatially and temporally continuous estimates of various hydrological processes acting as critical drivers of groundwater dynamics.

We agree with Reviewer 2 that descriptions of static features in table 2 are rather brief for readers who are unfamiliar with this data. We will add extended descriptions of the static features and the associated uncertainty in the revised manuscript.

In table 2, self-derived means that we did not take these static features from an existing dataset, but we calculated them from the dynamic meteorological input features ourselves. We will clarify this in the table, thank you for highlighting the ambiguity.

Yes, additional dynamic input features such as snow water equivalent, soil moisture or others have the potential to positively impact model performance. However, the overall best performance was not the scope of this study, which deals with theoretical considerations regarding entity awareness.

We relied on the HYRAS dataset, as it has proved its suitability in several studies before (e.g. Wunsch et al. 2021, 2022), and has a higher spatial resolution than the global datasets available. Moreover, we used the same meteorological dataset as in Wunsch et al. (2022) allowing a better comparison with their results for the single well method.

We will sharpen the formulation of the research aims in the introduction, and the data section regarding HYRAS in order to make this point clearer.

The authors should include the model forecasts beyond January 2016. While it may be challenging to obtain in-situ groundwater levels between 2016-present, it would be interesting to observe how the model predictions compare to the GRACE- and GRACE Follow-On (GRACEFO)-based total water storage changes (https://grace.jpl.nasa.gov/data/data-analysis-tool/) at a regional or national scale. This comparison would serve as an additional model validation and strengthen the manuscript.

We agree that inclusion of model forecasts beyond 2016 would be advantageous. However, because we rely on a previously published dataset, we have no way of updating this data.

Also, we argue that GRACE could not feasibly be used as a substitute for groundwater level measurements or as a comparison within the scope of our study, due to its inherent coarse spatial and temporal (monthly) resolution. Moreover, it constitutes a different variable (total water storage changes, as opposed to groundwater level directly) with inherent uncertainties in the computation of groundwater storage and subsequently groundwater levels, relying on various additional data. This would distort the original scope of our study, which focuses on theoretical and methodological considerations regarding entity awareness.

There should be an additional section (or a subsection within the Introduction) describing the study area and related studies on groundwater level changes. Also, the spatial distribution of the 108 well locations should be shown on a map.

We agree with Reviewer 2, that a (sub-)section describing study area, accompanied by a map, will be useful. We will include it.

Regarding a (sub-)section on 'related studies on groundwater level changes', it is unclear what reviewer 2 means by this broad formulation. There are multiple entire research fields occupied with groundwater level changes. We are confident that we reviewed the literature relevant to our study's domain allocation appropriately in the introduction, but will happily update this discussion with the resourced provided by Reviewer 2 (e.g. the ones provided in Reviewer 2's final major comment).

What are the 11 land cover classes in the CLC data? How are these used in the model? Can the categories be reduced by aggregating to a base class? E.g., crops aggregated to 'Agriculture,' urban/industry to 'Urban,' and so on? Is there no significant change in built-up or irrigated areas within the temporal domain of the model? The potential effects of land use changes on the model performance should be discussed. Also, the percentage of land use classes should be described in the Study Area section.

> The CLC is used as a one-hot-encoded static feature input to the ENVfeat model variant, next to the other 17 static environmental features. Yes, they could be reduced to fewer classes, there are e.g. 3 forest classes that could be combined into one, and 4 different urban classes that could be combined. However, from our conceptional understanding, using the single classes as defined makes more sense (e.g. the groundwater recharge in coniferous forest is significantly smaller than in deciduous forest, thus groundwater level should react differently in both forest classes to meteorological inputs, the same applies to continuous and discontinuous urban fabric etc.).

> Second, yes, there can be land use change over time, but in general, Germany – as a highly developed and densely populated country where use of land is subjugated to densely layered interests with extensive laws preventing unauthorized land use change – has very limited land use change over time. There is one notable exception, namely that about 8% of the countries area switched from arable land to forests over the period 1982-2016 (Song, 2018). All other land use types remained stable. We therefore consider land-use as a quasi-static feature. We will add text to explaining this aspect in the revised version of the manuscript.

The corresponding time series of the dynamic predictors for the two wells in Figure 5 should be added and tied up with the discussion related to the permutation feature importance.

> Thank you for this suggestion. We could add a figure that includes the dynamic predictors (at least P, T and rH) for the shown wells. However, we think that this would overload figure 5 at the present state and distract the reader from the actual point of discussion in figure 5 (i.e. the comparison of the in-sample and out-of-sample performance). If you insist, we suggest adding a separate figure. However, we think that setting the dynamic inputs features for selected single wells into relation to the permutation feature importance, which shows an average importance over all wells, is difficult anyway.

Evapotranspiration (ET) is the second largest component of the water cycle after precipitation (https://openetdata.org/what-is-evapotranspiration) and is a critical driver of groundwater use, which, in turn, is correlated to groundwater levels (Majumdar et al., 2020; 2022; Brookfield et al., 2023; Melton et al., 2021; Senay et al., 2022). Why didn't the authors include it as a dynamic predictor and instead rely on the potential ET (Table 2)? While the OpenET and the Landsat-derived actual ET products are currently available only over the conterminous United States (CONUS), the globally available 500 m MOD16 actual ET is available within the temporal domain of the model. Thus, the authors should justify the choice of their predictors.

> We agree that ET is an important dynamic predictor for groundwater levels, which could potentially improve the overall model performance. But as pointed out already above, we stick to the HYRAS dataset for the dynamic meteorological inputs (which does not include ET as a modelled parameter) for several good reasons. Moreover, ET is mainly controlled by temperature and relative humidity, which are included in our dynamic predictors.

Lines 50-60: While the proposed machine learning-based method of using multiple wells to develop an entity-aware global groundwater level prediction model is new, earlier studies have integrated

remote sensing, climate, and hydrogeologic data in a machine learning framework for estimating annual groundwater withdrawals (Majumdar et al., 2020; 2021; 2022; Wei et al., 2022) and land subsidence (Smith & Majumdar, 2020; Hasan et al., 2023). For the studies on groundwater withdrawal estimation, a single machine learning model was trained and validated using in-situ pumping measurements from multiple wells across vast geographical areas (states of Kansas and Arizona in the U.S.). Thus, the authors should clearly convey that the novelty lies in groundwater level monitoring rather than the entire hydrogeology domain.

> We thank Reviewer 2 for highlighting this research. We were not aware of these studies and will include them in the introduction section where existing global models are discussed (line 50 ff.), and point to the novelty concerning groundwater level modelling.

**Minor Comments**

Line 94- Fix typo: followed by a brief *introduction.*

> We will fix this, thank you.

What are the spatial resolutions of the predictor data listed in Table 2? How do the authors map the groundwater wells to these gridded raster datasets?

> Thanks for spotting this. We forgot to add the information that environmental feature values were simply selected at the location of the respective groundwater well. We will add this information and will specify resolution in Table 2.

Report other error metrics like the coefficient of determination (R2), root mean square error (RMSE), and the mean absolute error (MAE) in Table 3.

> Yes, we can report other error metrics. We will add R^2 and KGE in table 3 (Reviewer 2 probably means table 3 here). RMSE and MAE are not suitable, because they are not comparable between different sites due to different reference height, amplitude etc.

The CLC acronym is not defined.

> Thank you for pointing this out. CLC first appears in Table 2, we will add definition there.

Do the authors scale all the features? What scaling is applied?

> As mentioned in line 187, all dynamic features were standardized, i.e. standard scaled. However, we realized that the scaling method for the static features is not mentioned in the paper. Thank you for pointing this out, we will add this information in section 2.2 or 3.1. Numerical static features were standard scaled as well, categoric static features were one-hot encoded.

Lines 235-240: For the out-of-sample setting, are the scores only calculated for the testing period of a well that has been left out of model training? Why not calculate the score for the entire period?

> Yes indeed, scores are only calculated for the testing period of each well. We will point this out in the revised version. From the aspect of data leakage, it would have been possible to include the scores for the entire period of the left-out wells in the out-of-sample setting. However, we decided to stick with the same test period to allow direct comparability with the scores in the in-sample setting. As reviewer #1has correctly remarked, there is a (for groundwater level time series practically unavoidable) bias in the test set, thus, changing it would mean to lose this comparability.

References:

Song, X.P., Hansen, M.C., Stehman, S.V., Potapov, P.V., Tyukavina, A., Vermote, E.F. and Townshend, J.R., (2018). Global land change from 1982 to 2016. Nature, 560(7720), pp.639-643. https://doi.org/10.1038/s41586-018-0411-9

Wunsch, A., Liesch, T., & Broda, S. (2021). Groundwater level forecasting with artificial neural networks: a comparison of long short-term memory (LSTM), convolutional neural networks (CNNs), and non-linear autoregressive networks with exogenous input (NARX). Hydrology and Earth System Sciences, 25(3), 1671-1687, https://doi.org/10.5194/hess-25-1671-2021

Wunsch, A., Liesch, T. and Broda, S. (2022) 'Deep learning shows declining groundwater levels in Germany until 2100 due to climate change', *Nature Communications*, 13(1), p. 1221. Available at: https://doi.org/10.1038/s41467-022-28770-2